# Transcriptome and Physiological Analysis of Rapeseed Tolerance to Post-Flowering Temperature Increase

**DOI:** 10.3390/ijms242115593

**Published:** 2023-10-26

**Authors:** Javier Canales, José F. Verdejo, Daniel F. Calderini

**Affiliations:** 1Institute of Biochemistry and Microbiology, Faculty of Sciences, Universidad Austral de Chile, Valdivia 5110566, Chile; 2ANID-Millennium Science Initiative Program-Millennium Institute for Integrative Biology (iBio), Santiago 8331150, Chile; 3Graduate School, Faculty of Agricultural Sciences, Universidad Austral de Chile, Valdivia 5110566, Chile; jose.verdejo@uach.cl; 4Plant Production and Plant Protection Institute, Faculty of Agricultural Sciences, Universidad Austral de Chile, Valdivia 5110566, Chile

**Keywords:** post-flowering temperature increase, *Brassica napus*, heat stress, transcriptome analysis, seed yield, seed number, seed weight, gene coexpression network analysis

## Abstract

Climate-change-induced temperature fluctuations pose a significant threat to crop production, particularly in the Southern Hemisphere. This study investigates the transcriptome and physiological responses of rapeseed to post-flowering temperature increases, providing valuable insights into the molecular mechanisms underlying rapeseed tolerance to heat stress. Two rapeseed genotypes, Lumen and Solar, were assessed under control and heat stress conditions in field experiments conducted in Valdivia, Chile. Results showed that seed yield and seed number were negatively affected by heat stress, with genotype-specific responses. Lumen exhibited an average of 9.3% seed yield reduction, whereas Solar showed a 28.7% reduction. RNA-seq analysis of siliques and seeds revealed tissue-specific responses to heat stress, with siliques being more sensitive to temperature stress. Hierarchical clustering analysis identified distinct gene clusters reflecting different aspects of heat stress adaptation in siliques, with a role for protein folding in maintaining silique development and seed quality under high-temperature conditions. In seeds, three distinct patterns of heat-responsive gene expression were observed, with genes involved in protein folding and response to heat showing genotype-specific expression. Gene coexpression network analysis revealed major modules for rapeseed yield and quality, as well as the trade-off between seed number and seed weight. Overall, this study contributes to understanding the molecular mechanisms underlying rapeseed tolerance to heat stress and can inform crop improvement strategies targeting yield optimization under changing environmental conditions.

## 1. Introduction

Climate-change-induced temperature fluctuations pose a significant threat to crop production, particularly in the Southern Hemisphere, where strong warming is predicted across various global warming scenarios [1,2,3,4]. Recently, Rivelli et al. (2021) reported a temperature increase in the Southern Cone of South America over the past 30–50 years, highlighting the urgency of understanding crop responses to these changes [5].

Southern Chile is a high-yield potential region for temperate crops, including rapeseed [6,7,8,9]. The increasing temperature due to climate change is a critical environmental factor affecting rapeseed, particularly during the grain filling period [10,11]. Although numerous studies have evaluated rapeseed responses to heat stress, most have been conducted under controlled conditions, limiting the applicability of their findings to field conditions [12].

Heat stress during the critical reproductive stages, such as flowering and seed filling, can result in significant losses in rapeseed seed yield [13]. High temperatures during flowering reduce pollen viability and fertilization success, leading to fewer seeds per pod [14]. Additionally, heat stress during seed filling shortens the duration of this stage, impairing seed development and lowering seed weights [15]. Both daytime and nighttime heat negatively impact reproductive development, with nighttime heat at flowering and seed filling substantially reducing the number of seeds per pod and overall seed yield [16].

Advancements in high-throughput RNA sequencing (RNA-seq) have illuminated the transcriptomic adaptations of rapeseed in response to heat stress [13]. Comparative analyses between rapeseed cultivars subjected to control and heat stress conditions have identified a conserved heat stress response mechanism, primarily mediated through the upregulation of heat shock proteins (HSPs) and downregulation of metabolic pathways [17]. Moreover, heat-tolerant and heat-sensitive cultivars display distinct stress-responsive transcriptional profiles, offering valuable insights into potential adaptive mechanisms [17].

Extending these insights, studies focusing on mature *B. napus* seeds have identified differentially expressed genes (DEGs) involved in protein processing, secondary metabolite pathways, and specific transcription factor families such as NAC, ERF, and bHLH [18,19]. These findings collectively elucidate the complex transcriptional landscape that contributes to compromised reproductive success under heat stress conditions.

The trade-off between seed number and seed weight is a critical factor in plant reproduction and crop yield improvement [20]. In rapeseed, seed weight is controlled by maternal genotype, which regulates seed size mainly through cell number [21]. The pod wall serves as a source of nutrients for seed development, with pod wall photosynthesis contributing to seed filling and final seed weight [21]. Transcriptome analysis of rapeseed has revealed genes involved in the interaction between seed weight (SW) and seed number (SN) [20], and genome-wide association studies have identified genetic loci controlling these traits [22].

Previous research on sensitivity of the seed yield to source-sink ratio reduction in southern Chile showed that rapeseed is more susceptible to source reduction between the start of flowering and 15 days after [8], which overlaps with the critical period of rapeseed [23]. Interestingly, seed yield showed partial or full resilience during the beginning of flowering from the start to 15 days after flowering (DAF) [8,23,24]. In a recent heat stress study conducted in Southern Chile [25], where air temperature increased after the beginning of flowering (from 10 to 20 DAF), seed weight of rapeseed was negatively affected, but not seed weight window. This background raises the question about the molecular mechanisms involved in the rapeseed response to both the increased temperature and source reduction during flowering and the beginning of grain filling.

The objective of this paper is to investigate the transcriptome and physiological responses of rapeseed to post-flowering temperature increases, providing valuable insights into the molecular mechanisms underlying rapeseed tolerance to heat stress. This knowledge can inform crop improvement strategies targeting yield optimization under changing environmental conditions.

## 2. Results

### 2.1. Effects of Heat Stress on Seed Yield and Quality Traits of Two Rapeseed Genotypes

The effects of heat stress on seed yield and quality traits of two rapeseed genotypes (Lumen and Solar) were evaluated in two experiments conducted under similar climate conditions during two consecutive growing seasons.

The heat stress treatment increased average temperature between 4.3 ${}^{\circ }$C and 4.8 ${}^{\circ }$C from the beginning of flowering to 15 days after flowering. The seed filling period was affected by heat stress across genotypes and seasons. For instance, seed filling period was shortened by three and six days in Lumen in Season 1 and 2, respectively, whereas Solar was delayed by one day in Season 1 and shortened by two days in Season 2.

Seed yield and seed number showed a significant interaction between genotype and heat stress (*p* = 0.03 and *p* = 0.04, respectively), indicating that both traits were negatively affected by heat stress but to different extents depending on the genotype (Figure 1). Lumen showed an averaged seed yield reduction of 9.3% and no significant change was found in seed number under heat stress, whereas Solar showed an average seed yield reduction of 28.7%, completely explained by the seed number reduction. Moreover, seed yield was negatively correlated with average temperature during the period of heat stress in Solar (R2 = 0.67; *p* = 0.05), but not in Lumen (*p* > 0.05).

Seed weight was affected by genotype (*p* = 0.03), with Lumen having higher seed weight than Solar. On the contrary, this trait did not show any significant response to heat stress (Figure 1). Seed oil concentration was also influenced by genotype (*p* < 0.001), with Lumen reaching higher values (50.5%) than Solar (48.7%). In addition, seed protein concentration was sensitive to both genotype (*p* < 0.001) and heat stress (*p* = 0.038). Contrary to oil, seed protein concentration was higher in Solar (17.0%) than in Lumen (15.4%) in control plants (Figure 1), supporting a trade-off, which has been often reported between these quality traits in rapeseed. Under heat stress, protein seed concentration increased in Solar by 13.1% only during the second growing season.

### 2.2. Multivariate Analysis of RNA-Seq Data Reveals
Differential Effect of Heat Stress on Source and Sink Tissues

To gain further insights into the molecular factors underlying heat stress response in rapeseed plants under field conditions, we conducted a multivariate RNA-seq analysis using pod and seed samples to distinguish the effects of stress on source and sink tissues. Our dataset incorporated samples from two independent seasons, with three biological replicates per condition during the first season and four biological replicates for the second season. In total, 112 samples were sequenced, generating 2500 million paired-end reads and averaging 22.3 million raw reads per sample, with a mean pseudoalignment of 16.4 million reads per sample (Appendix A).

The multifactorial analysis revealed minimal to non-existent interaction between 2, 3, or 4 factors in both tissues, with most transcriptomic changes being attributable to the primary factors (Figure 2A,B). Notably, genotype was the predominant factor, affecting 28,012 and 29,091 genes in siliques and seeds, respectively (Figure 2A and Figure 2B). Interestingly, the number of genes affected by other factors was also of similar magnitude in both tissues, except for heat treatment. We found 8.5 times more heat-regulated genes in siliques than in seeds (1698 vs. 202 genes, q-value < 0.01), suggesting that source tissues are considerably more sensitive to temperature stress. Moreover, the overlap between these two sets of heat-regulated genes is only 35 genes (Appendix A), indicating a tissue-specific response to this factor.

### 2.3. Hierarchical Clustering Unveils Patterns of Heat-Induced Gene Expression in Siliques

To investigate how heat-responsive genes are expressed in siliques, we performed a hierarchical clustering analysis on the 1698 genes that were significantly impacted by temperature (q-value < 0.01) in these maternal tissues. We used the “Dynamic Tree Cut” package in R to determine the optimal number of clusters, which was eight (Figure 3A). These clusters showed different patterns of gene expression in response to heat treatment and between genotypes.

Clusters 1, 2, and 5 comprised genes that were downregulated by heat stress, representing around 60% of the total heat-responsive genes (947 out of 1698 genes). Most of these genes had higher expression levels in Lumen than in Solar (Figure 3A). Cluster 1 contained genes that were more highly expressed at 7 DAF than at 14 DAF, and were enriched for biological processes related to photosynthesis and starch metabolism. Cluster 2 included genes that were associated with pigment metabolism and response to light. Cluster 5 had no significantly enriched biological process (Figure 3B).

Clusters 3, 4, 6, 7, and 8 consisted of genes that were upregulated by heat stress, but they were smaller (between 80 and 200 genes) and less functionally enriched than the downregulated clusters. Clusters 3 and 7 had no significantly enriched biological process (Figure 3B). Clusters 4 and 6 showed consistent differences in gene expression between genotypes, with higher levels in Solar than in Lumen. These clusters were enriched for biological processes involved in cell wall metabolism, carbohydrate metabolism, and protein folding (Figure 3B).

Cluster 8 was the most interesting cluster, as it was highly enriched for genes involved in protein folding and showed a genotype-specific temporal response to heat stress. At 7 DAF, the expression levels and heat response of these genes were similar between genotypes. However, at 14 DAF, these genes exhibited a strong decrease in expression in Lumen but not in Solar (Figure 3B).

Taking together, these findings suggest that heat treatment primarily has a negative impact on the expression of genes associated with photosynthesis in siliques. The results also highlight distinct gene clusters that reflect different aspects of heat stress adaptation in this tissue. In particular, the biological functions associated with cluster 8 genes suggest a role for protein folding in maintaining silique development and seed quality under high temperature conditions.

### 2.4. Hierarchical Clustering Reveals Three Distinct Patterns of Heat-Responsive Gene Expression in Seeds

In the case of seeds, the hierarchical clustering analysis on 202 heat-responsive genes revealed three clusters with distinct expression patterns (Figure 4A). Cluster 1 (81 genes) was negatively regulated by temperature, cluster 2 (63 genes) was positively regulated by temperature, and cluster 3 (58 genes) showed differential expression between the genotypes and positive response to heat treatment (Figure 4A).

Cluster 1 and 2 also exhibited a temporal effect on their expression levels. In cluster 1, most genes had lower expression at 14 DAF than at 7 DAF or 21 DAF, regardless of the temperature. In contrast, in cluster 2, most genes had higher expression at 14 DAF than at other time points, regardless of the temperature (Figure 4A). Gene ontology analysis indicated that cluster 1 genes were involved in “glycolipid metabolism” and “cellular response to phosphate starvation”, whereas cluster 2 genes were involved in “carboxylic acid metabolic process” and “sucrose biosynthesis” (Figure 4B).

Cluster 3 was the only cluster that showed a clear genotypic difference in gene expression. However, this difference was not consistent across time or seasons (Figure 4A). For instance, at 7 DAF in the first season, the genes in this cluster had lower expression in Lumen than in Solar, but this trend was reversed at the same time point in the second season. Moreover, gene ontology analysis revealed that this cluster was enriched in genes related to “protein folding” and “response to heat” (Figure 4B).

### 2.5. Genotype-Specific Effects of Heat Stress on Photosynthesis-Related Genes in Siliques

Our physiological analysis revealed that the Lumen genotype exhibits increased resistance to heat stress compared to Solar in terms of seed yield. To investigate the genetic basis of this differential tolerance, we used the GeneOverlap R package to identify genes that were regulated by both genotype and heat stress in each group of tissues.

We found a significant intersection of 1015 genes in siliques (*p*-value = 5 × 10${}^{-36}$, Figure 5A), but not in seeds, suggesting that siliques are more sensitive to genotype-dependent heat responses. We then mapped these 1015 genes to the clusters of siliques shown in Figure 3 and found that most of them belonged to clusters 1 and 2 (Figure 5B), which were enriched for photosynthesis and light response genes and showed a negative response to heat stress (Figure 3A).

Moreover, these clusters consistently exhibited higher expression levels of genes in Lumen than in Solar, indicating that Lumen maintains photosynthetic activity better than Solar under heat stress. To illustrate these differences, we selected two representative genes from each cluster: *SBPase* from cluster 1, which encodes the chloroplast enzyme sedoheptulose-1,7-bisphosphatase involved in the carbon reduction of the Calvin cycle [27], and *INAP1* from cluster 2, which encodes the catalytic subunit of the chloroplastic ferredoxin/thioredoxin reductase [28]. In both cases, we observed significant differences between genotypes in most of the samples analyzed, showing lower mRNA levels in Solar than in Lumen (Figure 5C,D).

### 2.6. Co-Expression Network Analysis Reveals Major Modules for Rapeseed Yield and Quality

To identify candidate genes related to rapeseed yield and quality, we performed a gene co-expression network analysis using the transcriptomes of siliques and seeds by the WGCNA method [29] as indicated in the materials and methods section. We constructed four co-expression networks corresponding to two different times after flowering (7 and 14 days) and two group of tissues (seeds and siliques) and compared them to select the network that showed the highest number of genes correlated with the agronomic parameters of yield and seed quality.

We found that the seed network at 7 DAF had the highest number of genes significantly correlated with the main agronomic traits of yield and quality (Appendix A). The seed network at 7 DAF showed more than five times as many genes correlated with seed number and yield than the silique networks at both 7 and 14 DAF (Appendix A). These results indicate that the co-expression network of seeds at 7 DAF is the most suitable for finding genes that explain the parameters of yield and seed quality.

The co-expression network of seeds at 7 DAF consisted of 35,432 genes distributed in 33 modules (Figure 6A). These traits included growth and development (plant height, branch per plant, siliques per plant, seeds per silique, weight per silique, biomass, and harvest index), seed quality (oil and protein content) and yield (seed number, thousand seed weight, yield). We identified eight co-expression modules with high correlation with some of these agronomic parameters (at least one absolute correlation of 0.7 or more and a *p*-value less than 0.0001, Figure 6B). Interestingly, five of these eight modules were correlated with quality and yield parameters and were also the largest ones in the network.

For seed quality, we found two modules positively correlated with protein content (the lightblue3 and firebrick modules) and one module positively correlated with oil content (lightskyblue2, Figure 6B). The overrepresentation analysis of biological functions showed that the lightblue3 and firebrick modules were enriched in genes related to protein synthesis and transport as well as photosynthesis and small molecule metabolism (Appendix A). The lightskyblue2 module was enriched in genes related to glycolysis, purine metabolism, ribonucleoprotein complex biogenesis, protein folding, and mRNA splicing (Appendix A).

### 2.7. Identification of Regulatory Factors Associated with the Compensation of SW and SN Flowering

In the case of yield, we found two highly correlated co-expression modules, “indianred2” and “palevioletred” with 3584 and 216 genes, respectively. Interestingly, both modules showed an inverse correlation between seed number and weight, suggesting that they could be related to the trade-off between these two yield-determining factors. Given the relevance of this topic for crop performance and improvement, we analyzed these modules in more detail.

As shown in Figure 7A, the genes in the indianred2 module showed a positive correlation with seed number and a negative correlation with weight. A biological function enrichment analysis revealed that it is especially enriched in genes associated with translation, as well as some functions related to transcription, protein ubiquitination and protein targeting (Figure 7B). To identify the regulators that control the expression of this module, we performed a regulatory network analysis using the R package Bionero and the transcription factor annotation from PlantTFDB 4.0 [30].

In this way, we inferred a network with 93 transcription factors and 1256 targets, being the five transcription factors with the highest connectivity *BnaC03g43430D* (*PHL2*), *BnaC05g18750D* (*INO1*), *BnaA03g18970D* (*ABI4*), *BnaC08g31840D* (*MYC67*), and *BnaC03g57540D* (*SEP3*) (Figure 7C). Interestingly, functional information available in Arabidopsis indicates that four of these five factors are related to floral architecture [31] or seed development [32,33,34].

In the case of the palevioletred module, the genes showed higher expression levels in samples with lower seed number and higher weight (Figure 8A). According to gene ontology analysis, the genes in this module are associated with biological functions related to intracellular transport, mRNA metabolism, cell signaling, and chromatin remodeling (Figure 8B). The regulatory network analysis revealed that this co-expression module is controlled by 118 TFs that potentially regulate 1483 targets. The five transcription factors with the highest connectivity were *BnaA06g29050D*, *BnaA04g24200D* (*OSX2*), *BnaC05g51140D*, *BnaC03g72370D* (*NGA1*) and *BnaA06g21460D* (*ARF2*) (Figure 8C). Importantly, the transcription factor ARF2 has been shown to increase seed size due to increased cell proliferation, suggesting that the regulators identified could be important for controlling seed weight [35].

## 3. Discussion

Brassica crops such as canola, mustard, and cabbage are cool-season plants that are susceptible to high temperatures, especially during reproductive development [14,36,37]. Even short episodes of heat stress during flowering can significantly reduce seed and oil yields in canola by impairing pollen viability, pollen tube growth, fertilization success, and embryo development [14,38]. Our study found genotype-specific responses to heat stress during flowering and early seed development. The Solar genotype showed greater reductions in seed yield and number compared to Lumen under heat stress. Seed oil and protein concentrations also showed genotype-dependent responses. Lumen maintained higher seed oil content whereas Solar increased protein content under heat stress. These results demonstrate differential heat tolerance between genotypes that impacts yield and quality; however, the underlying molecular mechanisms are unknown. This motivated a large-scale transcriptomic analysis (112 samples) which provides novel insights into the molecular mechanisms governing heat stress responses in rapeseed during the critical reproductive stages of flowering and early seed set. Leveraging field experiments and multi-level omics across source and sink tissues, we reveal both conserved and divergent strategies for thermal adaptation.

### 3.1. Divergent Molecular Responses to Heat Stress in Source and Sink Tissues

Our multivariate analysis of RNA-seq data revealed distinct effects of heat stress on siliques (source) versus seeds (sink). In the pod wall, heat stress triggered extensive transcriptional repression of genes regulating photosynthesis and carbon assimilation. This corroborates previous observations in model species like Arabidopsis, where high temperature inhibits RuBisCO activase [39], electron transport, and carbon fixation [40]. Exposure to high temperatures was found to negatively impact components of the photosynthetic machinery in *B. napus* seeds, including reduced oil accumulation, lowered photosynthetic and respiration rates, decreased maximum quantum yield of photosystem II, and reduced levels of light-harvesting pigments like chlorophyll [41]. Moreover, a previous transcriptomic analysis in *B. napus* seeds also revealed up-regulation of genes related to the response to high light intensity and genes associated with chloroplasts and photoinhibition in response to heat stress [41].

In maturing seeds, the predominant heat response entailed upregulation of protein homeostasis processes like folding and stabilization. Such activation of heat shock factors, chaperones, and related pathways enables survival under protein-denaturing conditions [42]. Intriguingly, Lumen exhibited fluctuating induction of these genes whereas Solar showed constitutively high expression. This hints at distinct regulatory logic where Lumen employs a reactive mechanism triggered by stress, whereas Solar utilizes an anticipatory strategy.

Our finding that photosynthesis-related genes were downregulated in siliques but not seeds under short heat waves suggests reproductive tissues are more sensitive than vegetative tissues to high temperature damage during flowering. This aligns with previous work showing decreased expression of sucrose transporters like OsSUT1 in rice grains exposed to heat stress, which limits photoassimilate supply and grain filling [43]. Reduced photosynthate translocation is a common consequence of heat stress and a major cause of yield loss [44,45,46].

### 3.2. Differential Transcriptome Rapeseed Genotype Responses to Heat Stress

We newly uncover genotype-dependent variation in the extent of photosynthetic gene repression. The heat-tolerant Lumen maintained higher expression of light reaction and Calvin cycle transcripts compared to the sensitive Solar. Sustaining pod photosynthesis likely maintains carbon supply to developing seeds under heat stress, as reflected by Lumen’s yield stability (Figure 1). We found that an important gene of the Calvin cycle, *SBPase*, was downregulated under heat stress and consistently showed lower levels in the Solar genotype. Previous research has shown the importance of SBPase activity in photosynthesis and plant growth [47]. Specifically, transgenic plants with decreased SBPase activity have exhibited reduced rates of CO${}_{2}$ assimilation and carbon fixation [48,49]. This provides evidence that SBPase plays a key role in regulating carbon flow in the Calvin cycle. Furthermore, overexpressing SBPase has been demonstrated to increase photosynthetic capacity, carbon fixation, biomass accumulation, and starch and sucrose levels in tobacco [50,51]. Therefore, the downregulation of SBPase we observed under heat stress likely contributes to decreases in photosynthesis and growth in heat sensitive genotypes. Our findings suggest *SBPase* could be a useful target for genetic modification to improve heat tolerance by maintaining carbon assimilation and plant productivity under high temperatures.

Our results are consistent with the study by Jedlickova et al., 2023 in which photosynth- esis-related genes clustered in two co-expression modules showing higher expression in heat-sensitive cultivars DH12075 and Westar compared to heat-tolerant Topas after heat stress [17]. The authors suggest induction of these genes may indicate heat damage to photosystems in sensitive cultivars. Genes connected to photosynthesis and light-harvesting proteins were also up-regulated by heat specifically in seeds of sensitive cultivars [17]. The effects of heat stress on impairing photosynthesis and related gene expression likely contribute to reduced seed production and oil content in sensitive *B. napus* cultivars. Identifying and selecting for thermotolerance factors that protect photosynthesis may improve yield stability.

### 3.3. Distinct Regulators Mediate the Seed Number versus Weight Trade-Off

Beyond heat responses, our integrated omics approach discovered novel gene modules regulating the complex physiological trade-off between seed number and weight, an area of intense agronomic interest. The identification of the indianred2 and palevioletred modules that show an inverse correlation between seed number and weight suggests there are specific genetic networks controlling this trade-off in rapeseed. To further explore these networks, we analyzed the transcription factors with the highest connectivity in each module.

In the case of indianred2, the transcription factors *INO* (*INNER NO OUTER*) and ABI4 (ABA INSENSITIVE 4) stand out for their relationship with seed development [33,52]. The Arabidopsis *INO* gene encodes a YABBY family transcription factor that acts exclusively in regulating development of the ovule outer integument, with loss-of-function mutants lacking this structure [32,53]. We identified the ortholog of INO in *B. napus* as a hub of a gene coexpression module that showed a positive correlation with seed number per silique and a negative correlation with seed weight. This correlation is consistent with the specific role of INO in promoting outer integument growth. In wild-type Arabidopsis plants, the outer integument facilitates pollen tube guidance, fertilization, and supports embryo development [32]. Thus, loss of INO function leads to reduced seed number due to failed fertilization, while seeds that do form in heterozygous plants show increased weight and size, likely due to additional resources available from unfertilized ovules [54]. The central role of *INO* in regulating outer integument growth and its downstream effects on seed number and weight underlies its identification as a hub of the correlated gene coexpression module.

Our finding that *ABI4* expression positively correlates with seed number in *B. napus* is consistent with previous studies demonstrating a role for *ABI4* in promoting seed production. Specifically, Shu et al., 2016 [33] showed that constitutive overexpression of *ABI4* in *Arabidopsis thaliana* resulted in plants with a 20–26% reduction in total seed yield compared to wildtype. This decrease was attributed to a significant reduction in seed number, as the 1000-grain weight was not affected by ABI4 overexpression [33]. The reduction in seed number was likely due to the altered ABA/GA balance caused by ABI4-mediated upregulation of ABA biosynthesis genes like NCED6 and downregulation of GA catabolic genes like *GA2ox7* [33]. The resulting increase in ABA and decrease in GA probably impaired fertilization and seed development. Taken together, these results indicate that precise control of ABI4 expression is critical for proper seed yield, as both too little and too much ABI4 can negatively impact production by reducing seed number. Thus, our observed positive correlation between ABI4 and seed number in *B. napus* suggests this gene may play a similar role in regulating the ABA/GA balance and reproductive development in oilseed crop species. Further characterization of ABI4’s effects on hormone signaling and the seed setting process in *B. napus* will clarify its utility as a target for improving seed yield.

Conversely, in the palevioletred module, *ARF2*(*AUXIN RESPONSE FACTOR 2*) emerges as a high-connectivity transcription factor. *ARF2* is a pivotal regulator in auxin signaling and is implicated in controlling seed size through the modulation of cell expansion and endoreduplication [35]. Our gene co-expression analysis shows a negative correlation with seed number but a positive correlation with seed weight in *Brassica napus*. This result aligns with previous findings in Arabidopsis thaliana that loss-of-function mutations in ARF2 lead to increased seed number but decreased individual seed weight, whereas *ARF2* overexpression has the opposite effect of decreasing seed number but increasing seed size [35]. The conserved role for *ARF2* in promoting seed growth across distinct *Brassicaceae* species is likely due to its activity in stimulating cell proliferation in the integuments of developing ovules, as demonstrated in Arabidopsis [55]. The enlarged integuments generated more cells in the seed coat, allowing increased embryo growth and final seed size. Taken together, these data establish *ARF2* as a key maternal regulator of seed size in multiple Brassicaceae species, functioning to promote integument cell proliferation and limit seed number in order to increase the biomass allocated per seed. Modulating *ARF2* expression or activity provides a promising route to altering seed size and yield in oilseed crops like canola. These transcription factors may represent important nodes in the co-expression modules that modulate the trade-off between seed number and weight in rapeseed.

The co-expression modules associated with the seed number versus weight trade-off differed between this work and our previous work focused on shading treatments effects on seed yield [20]. In the present study under heat stress, the indianred2 module linked to higher seed number showed enrichment for translation and protein functions, whereas the palevioletred module for higher seed weight was associated with transport and signaling. In contrast, in our previous work assessing light stress [20], the turquoise module correlated with seed number was enriched for cell cycle and DNA replication genes, whereas the blue module for seed weight showed lipid metabolism functions. These results indicate distinct molecular processes mediate the trade-off in response to heat versus shading. The hub transcription factors also differed between studies. This work identified floral, embryo and growth regulators as key for seed number or weight (*INO*, *ABI4*, *ARF2*) whereas Canales et al., 2021 [20] highlighted *NAC082*, *NGAL2,* and *IDD1*. The distinct functional enrichment and hub genes likely reflect the different environmental stimuli used in each study to modify source-sink balance during seed set. The low overlap in the modules and regulators identified between the two studies indicates there are mostly specific, rather than conserved, molecular mechanisms mediating the trade-off in response to heat versus shading. Further comparative analyses to find common genetic factors influencing plasticity in seed number and weight would be valuable, but the current results suggest the underlying networks are largely distinct.

## 4. Materials and Methods

### 4.1. Field Experiment, Treatments, and Crop Management

A detailed description of the experimental setup, crop management, and weather data is provided in the Appendix A section. Briefly, two field experiments were set in the 2019–20 and 2020–21 growing seasons at the Austral Farming Experimental Station (EEAA) of the Universidad Austral de Chile in Valdivia, Chile (39${}^{\circ }$47${}^{\prime }$ S, 73${}^{\circ }$14${}^{\prime }$ W). In each experiment, two adapted spring rapeseed hybrids (Lumen and Solar from NPZ-Lembke, Germany) were assessed under two heat stress treatments (Appendix A): a control without manipulation and a thermal increase treatment where plots were heated by 5 ${}^{\circ }$C on average above the environmental temperature from the beginning of flowering [BBCH 61] to 15 days after flowering. The hybrids Lumen and Solar were chosen for their similar phenology and confirmed adaptation to southern Chile. The experiment of the 2019–20 growing season was arranged in a split plot design with three replicates in which the heat stress treatments were assigned to the main plots and genotypes to sub-plots. In the experiment of the 2020–21 growing season, plots were arranged in a randomized block design with three replicates in which the heat stress treatments and genotypes were fully randomized in each block. In both experiments, rapeseed plants were sown at a plant density of 55 plants m${}^{-2}$.

Air temperature and average solar radiation between emergence and the start of flowering recorded across the experiments were 10.9 and 11.4 ${}^{\circ }$C and 17.2 to 18.9 MJ m${}^{2}$ day${}^{-1}$, respectively (Appendix A). Crop phenology of control plants was similar between genotypes and seasons.

### 4.2. Phenology and Physiological Plant Sampling

Crop development was followed twice a week in both experiments according to the BBCH phenological scale for rapeseed [56]. In order to determine seed yield (SY), seed number (SN), thousand seed weight (TSW), and quality traits as grain oil and protein concentrations, the seed samples were harvested in one lineal meter from the central rows of each plot at maturity (BBCH 89), when the seeds inside the siliques were dark and hard. Seed yield and number was measured after oven drying the samples at 65 ${}^{\circ }$C for 48 h. Seed number was counted by using a grain counter (Pfeuffer GmbH, Kitzengen, Germany). Then, SY was measured and TSW was estimated as the ratio between seed yield and SN.

Oil concentration of seeds was determined by near infrared reflectometry (NIR) (Foss Infratec 1241, Hilleroed, Denmark) and nitrogen concentration of seed by the Kjeldahl procedure [57]. Seed protein concentration was calculated with a conversion factor of 5.8 [58]. Seed concentrations of both oil and protein were expressed on a dry matter basis.

### 4.3. Seed Sampling and RNA Isolation

Seed samples for RNA-seq analysis were collected from 5 plants per plot (25 siliques per plant from the basal portion of the primary inflorescence) across treatments at two developmental stages: 7 and 14 days after flowering, corresponding to one and two weeks after initiation of the heat stress treatment, respectively. The precise collection dates for each genotype are delineated in Appendix A. Immediately after harvest, the seed samples were flash frozen in liquid nitrogen and stored at −80 ${}^{\circ }$C until processing. The frozen seeds were ground into a fine powder using liquid nitrogen and a chilled mortar and pestle. Total RNA was extracted from 100 mg of each pulverized seed sample using a protocol adapted for Brassica seeds [59] and NucleoSpin Gel and PCR clean-up columns (Macherey-Nagel, Düren, Germany) as described previously [60].

### 4.4. RNA-Seq Analysis

The RNA-seq analysis incorporated biological replication, with samples collected from multiple field plots under each condition. In the first field trial season, 3 biological replicates were analyzed per genotype, tissue type, temperature treatment, and time point, for a total of 48 samples. In the second field season, 4 biological replicates were analyzed, yielding 64 samples. Each biological replicate represents plant material harvested from an independent field plot receiving the same conditions. The replicates reflect the inherent biological variability between plants grown under field conditions. RNA samples were transported to Novogene facilities located in Sacramento, CA, USA, under cryogenic conditions utilizing dry ice. Subsequent analyses for RNA quality, as well as library construction and high-throughput sequencing, were executed by Novogene’s Beijing-based laboratory. For each RNA sample, a unique Illumina RNA-seq library was constructed using the Illumina TruSeq Stranded Total RNA Sample Prep Kit according to the manufacturer’s protocol. Individual barcodes were incorporated during the library preparation for later demultiplexing. Sequencing was conducted employing a 2 × 150 bp paired-end sequencing kit on an Illumina NovaSeq 6000 platform, targeting a sequencing depth of 6 Gb per individual sample. Quality control metrics revealed an average Q20 score of 97.9% and a Q30 score of 94.0% across all samples, thereby attesting to the high fidelity of the sequencing process.

The RNA-seq reads were subjected to a comprehensive computational workflow utilizing pseudoalignment algorithms for accurate transcript quantification and downstream differential expression analysis. Specifically, the Kallisto software (version 0.46.1) was employed for the initial quantification steps [61]. A reference transcriptome index was generated using the “kallisto index” function and rapeseed annotation version AST_PRJEB5043_v1, subsequent to which the “kallisto quant” function was executed on each sample with 100 bootstraps with default parameters. The output provided transcript abundance estimates in terms of transcripts per million (TPM).

For the identification of differentially expressed genes (DEGs) between distinct experimental conditions, the R-based statistical package Sleuth (version 0.30.0) was employed [62]. A Sleuth object was instantiated via the “sleuth_prep” function, incorporating multiple variables into the model, namely genotype, heat treatment, temporal factors, and seasonal variations. Two statistical models were then fitted using the “sleuth_fit” function: a reduced model incorporating all main effects and their respective interactions, and a full model encompassing a four-way interaction term. Likelihood ratio tests (LRT) were executed between these models using the “sleuth_lrt” function, culminating in the identification of DEGs using an adjusted *p*-value < 0.01.

### 4.5. Hierarchical Clustering Analysis

For the purpose of uncovering the latent structure in gene expression profiles, hierarchical clustering was conducted. Initially, the gene expression matrix was subjected to normalization, centering, and scaling using native R functions. The Pearson correlation distance matrix was subsequently employed for hierarchical clustering through the “hclust” function, utilizing average linkage as the agglomeration method. The Dynamic Tree Cut algorithm, implemented in the “cutreeDynamic” function from the dynamicTreeCut package [63], was leveraged to partition the hierarchical tree into an optimal number of gene expression modules. Visualization of these modules was facilitated using the ComplexHeatmap package [26].

Functional enrichment analysis of the gene modules was performed using the g:Profiler toolkit [64]. The gene sets corresponding to each identified module were subjected to Gene Ontology (GO) enrichment analysis against *Brassica napus* gene databases using the “gost” function implemented in the R package gprofiler2 [65]. This enabled the elucidation of the biological pathways and molecular functions significantly associated with each gene module.

### 4.6. Gene Coexpression Network Analysis

To identify modules of coexpressed genes associated with seed yield and quality traits, we performed weighted gene coexpression network analysis (WGCNA) as described previously [29]. The analysis was conducted using the R package BioNERO [66] on normalized RNA-seq count data.

As a first step, we fitted a soft thresholding function to the count data to determine the power (β) for weighted network construction. Using the “SFT_fit function” with a signed hybrid network type, Pearson correlation, and a target R^2^ of 0.9 for scale independence, a power of 8 was selected. Next, the exponentiated transformed count data was used to construct a signed coexpression network with the “exp2gcn” function of BioNERO R package [66], specifying the previously determined soft threshold power, a Pearson correlation, and a module merging threshold of 0.8.

The resulting gene coexpression network consisted of 35,432 genes grouped into 33 modules based on topological overlap. To identify modules associated with phenotypes, we calculated module-trait correlations using the module_trait_cor function. Eight modules showed significant correlations with at least one trait related to plant growth, seed yield, or quality (|r| > 0.7 and *p* < 0.0001).

To visualize the module-trait relationships, a customized heatmap was generated using the ComplexHeatmap package in R [26]. Modules were assigned to three broad categories (growth, yield, quality) and ordered accordingly. The heatmap depicts module–trait correlations, with the strength of positive correlations shown in red and negative correlations in blue. This systems-level visualization reveals major gene coexpression modules associated with different yield and quality traits.

To infer potential regulators of the yield and quality associated modules, we performed gene regulatory network analysis using the “exp2grn” function of BioNERO R package [66]. Transcription factors were specified based on PlantTFDB 4.0 annotations [30], and the expression data was used to predict regulatory interactions [66]. Hub regulators with the highest connectivity in each module were identified using the “get_hubs_grn” function of BioNERO R package [66], which implements three popular algorithms: GENIE3 [67], ARACNE [68], and CLR [69]. The transcriptomic network analysis provides a systems-level view of potential genetic factors influencing yield and quality traits in response to heat stress.

## 5. Conclusions

In summary, our multi-level study of rapeseed under field conditions provides new insights into heat stress responses during the critical reproductive phase. We reveal genotype-dependent variation in photosynthetic gene regulation that may confer thermotolerance through sustained carbon assimilation. Our integrated omics approach also newly identified candidate networks and transcriptional regulators governing the agronomically important trade-off between seed number and weight. These findings advance molecular understanding of heat adaptation and yield determination in oilseed crops.

## Figures and Tables

**Figure 1 ijms-24-15593-f001:**
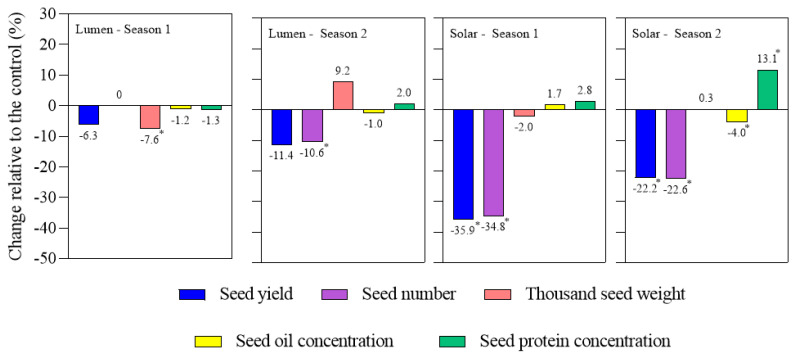
Response of seed yield, seed number, thousand seed weight, and quality traits (seed oil and protein concentrations) across seasons due to heat stress relative to the control. Asterisks indicate significant effect of heat stress treatment on the control by using Fisher’s LSD test (*p* < 0.05).

**Figure 2 ijms-24-15593-f002:**
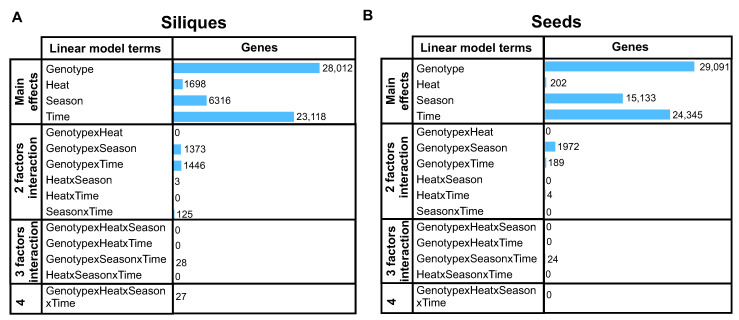
Multivariate analysis of RNA-seq data in siliques and seeds under heat stress. The bar plots display the number of differentially expressed genes (DEGs) attributed to various factors and their interactions in (**A**) siliques and (**B**) seeds based on sleuth analysis (q < 0.01). Genotype is the predominant factor affecting gene expression, whereas heat treatment has a more pronounced effect on siliques than seeds, indicating tissue-specific responses to temperature stress. Data were obtained from field experiments conducted in Valdivia, Chile, using two rapeseed genotypes (Lumen and Solar) subjected to control and heat stress conditions.

**Figure 3 ijms-24-15593-f003:**
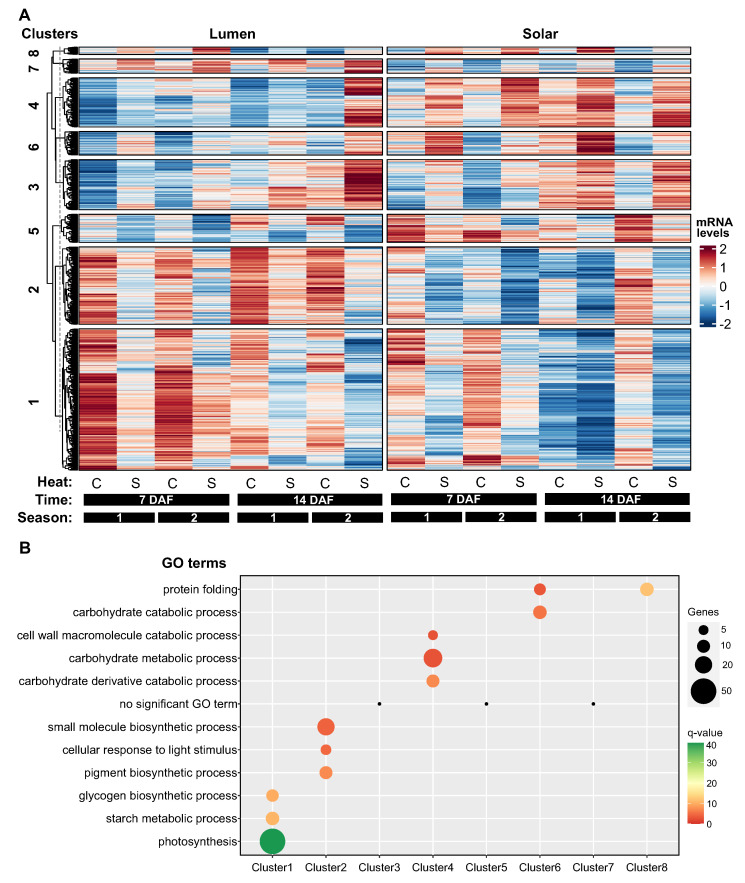
Hierarchical clustering analysis reveals distinct patterns of heat-responsive gene expression in siliques. (**A**) Heatmap showing hierarchical clustering of 1698 heat-responsive genes in siliques into eight distinct clusters based on their expression profiles across samples using ComplexHeatmap [26]. Rows represent genes and columns represent samples from two genotypes (Lumen and Solar), two heat treatments (control and heat stress), two time points (7 and 14 DAF), and two seasons. (**B**) Enrichment analysis of biological processes for each gene cluster. The number of genes and FDR-adjusted *p*-values are shown for the top enriched terms in each cluster. The black dots positioned above Clusters 3, 5, and 7 denote an absence of statistically significant GO terms. Clusters 1, 2, and 5 contain genes downregulated by heat stress and involved in photosynthesis, pigment metabolism, and carbohydrate metabolism. Clusters 3, 4, 6, 7, and 8 show heat-induced genes associated with various processes including protein folding and cell wall metabolism. Cluster 8 displays a genotype-specific temporal response, with decreased expression of protein folding genes in Lumen but not Solar at 14 DAF under heat stress.

**Figure 4 ijms-24-15593-f004:**
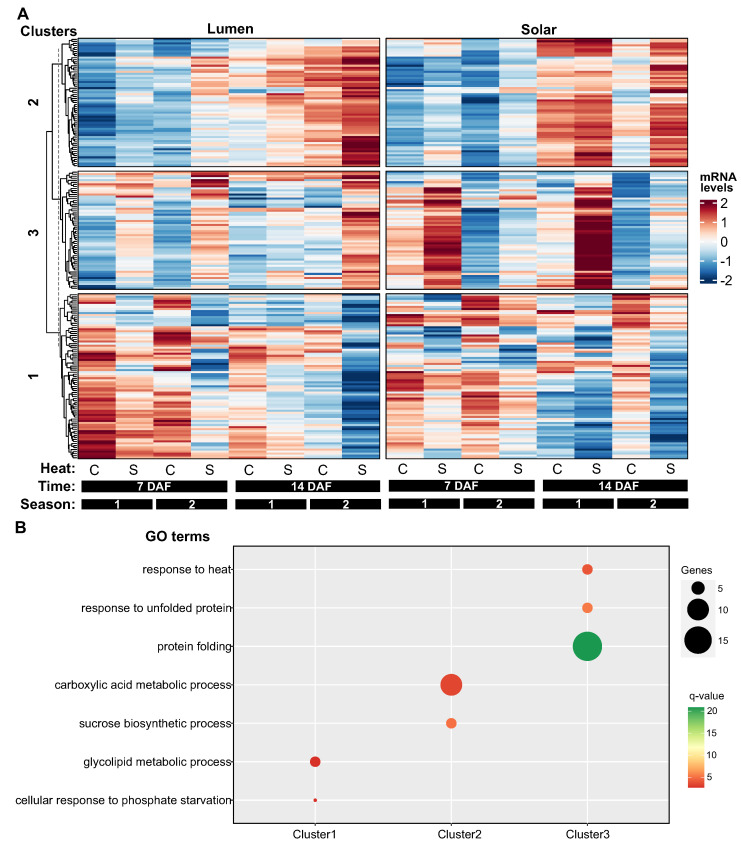
Hierarchical clustering reveals three distinct patterns of heat-responsive gene expression in rapeseed seeds. (**A**) Heatmap visualization of 202 differentially expressed genes (q-value < 0.01) in response to heat treatment, clustered into three groups with distinct expression profiles. Rows represent genes and columns represent different genotype (Lumen or Solar), timepoint (7 or 14 days after flowering), season (1 or 2), and temperature treatment (control or heat stress) conditions. Normalized expression values are color-coded based on row z-score. Cluster 1 (81 genes) shows downregulation by heat stress. Cluster 2 (63 genes) exhibits upregulation by heat. Cluster 3 (58 genes) displays genotype-specific responses to heat. (**B**) Gene ontology enrichment analysis indicates Cluster 1 is involved in glycolipid metabolism and phosphate starvation response, Cluster 2 in carboxylic acid metabolism and sucrose biosynthesis, and Cluster 3 in protein folding and response to heat. The small dot positioned above Cluster 1 denote an absence of statistically significant GO terms.

**Figure 5 ijms-24-15593-f005:**
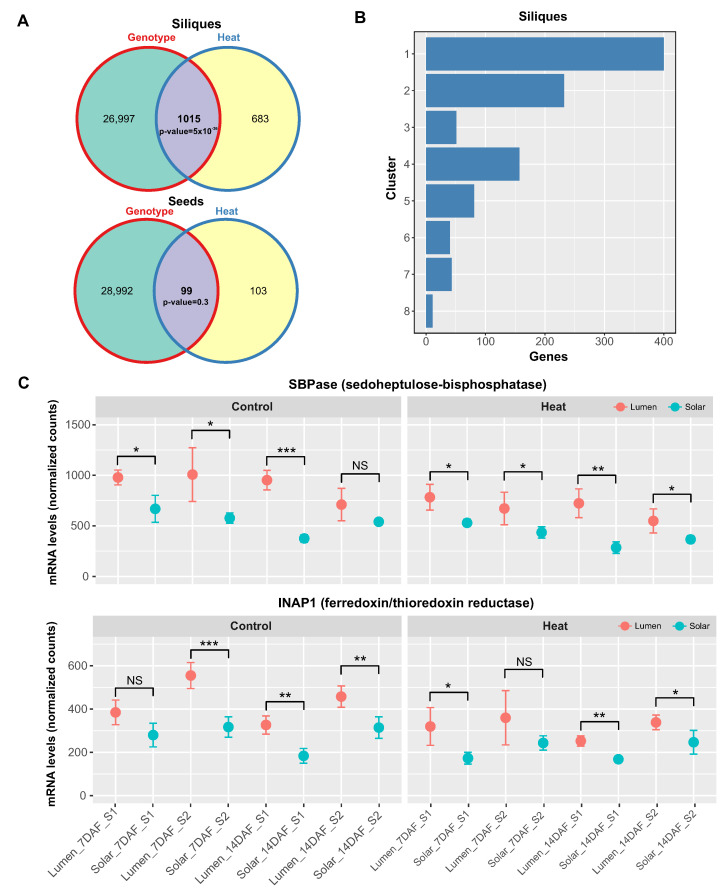
Genotype-dependent effects of heat stress on photosynthesis-related genes in siliques. (**A**) Venn diagrams showing the number of genes significantly regulated by heat, genotype, or both factors in siliques and seeds. Only siliques showed a significant overlap between the two factors (*p* < 0.05, GeneOverlap test). (**B**) Histogram showing the distribution of the 121 genes that were regulated by heat and genotype in siliques across clusters 1 and 2, which contain photosynthesis and light response genes. (**C**) Expression profiles of the photosynthesis-related genes *SBPase* (cluster 1) and *INAP1* (cluster 2) in silique samples under control and heat conditions. *SBPase* and *INAP1* mRNA levels were consistently higher in the Lumen genotype compared to Solar under heat stress based on RNA-seq data, suggesting improved maintenance of photosynthetic gene expression in the heat tolerant Lumen variety. Bars represent means ± SE (n = 3). Asterisks indicate significant differences between genotypes (NS, not significant, * *p* < 0.05, ** *p* < 0.01, *** *p* < 0.001).

**Figure 6 ijms-24-15593-f006:**
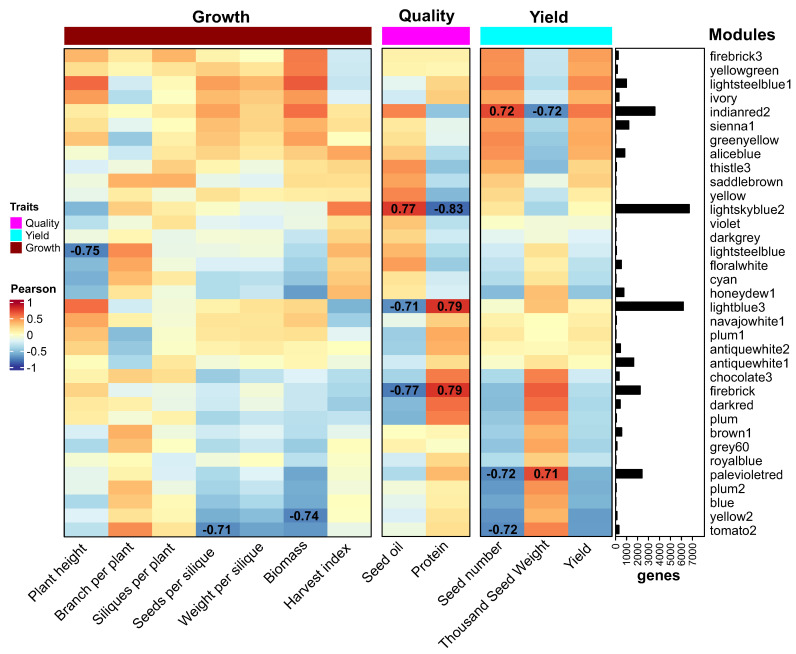
Correlation heatmap of gene coexpression modules with yield and quality traits in rapeseed. Heatmap showing Pearson correlation coefficients between module eigengenes and phenotypic traits related to plant growth, seed yield, and quality. The modules were generated by weighted gene coexpression network analysis of transcriptomic data from 28 samples of seeds at 7 days after flowering. Modules are grouped into categories and ordered based on strongest correlations. Red indicates a positive correlation, blue indicates a negative correlation. The largest modules positively correlated with protein content (lightblue3, firebrick), oil content (lightskyblue2), seed number (indianred2), and seed weight (palevioletred) are highlighted. This systems-level analysis reveals major coexpression modules associated with key agronomic traits that determine yield and quality in rapeseed. The module names (e.g., lightblue3, firebrick, indianred2) correspond to coexpression modules identified by the WGCNA algorithm.

**Figure 7 ijms-24-15593-f007:**
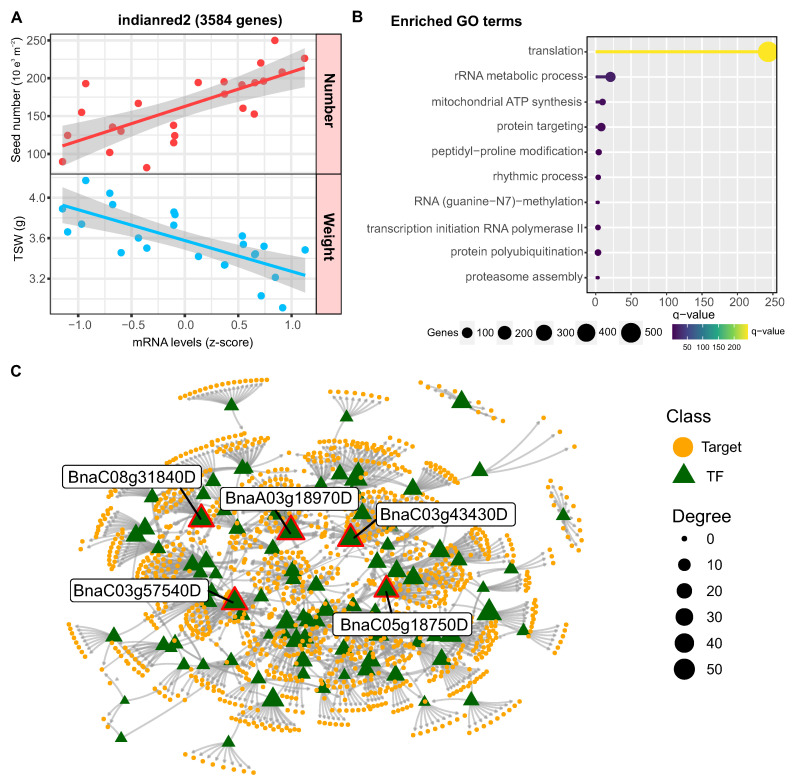
Co-expression network analysis reveals regulators of seed number and weight tradeoff. (**A**) Correlation plot showing the relationship between average expression of indianred2 module genes and seed number/weight across samples. Each point represents the mean expression level of module genes for a sample, with red indicating seed number and blue indicating seed weight. (**B**) Enriched Gene Ontology terms for the indianred2 module related to translation, protein targeting, and other functions. The x-axis shows statistical significance. GO term enrichment analysis was performed using g:Profiler with a q-value cutoff of 0.05 (**C**) Inferred gene regulatory network for the indianred2 module. The network was constructed using GENIE3, ARACNE, and CLR algorithms on gene expression data. Nodes represent transcription factors (triangles) and target genes (circles). The 5 transcription factors with highest connectivity (degree) are labeled. Node size corresponds to degree.

**Figure 8 ijms-24-15593-f008:**
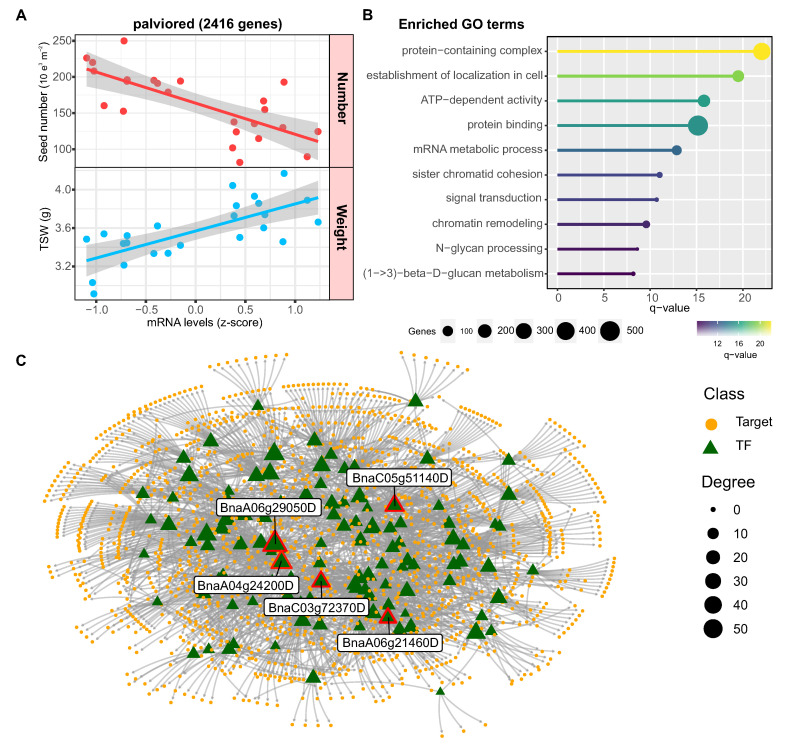
Co-expression module associated with seed weight in rapeseed. (**A**) Scatterplot showing the negative correlation between seed number and the expression pattern of genes in the palevioletred module (r = −0.77, *p* < 0.0001). Each dot represents a specific sample. (**B**) Enriched gene ontology (GO) terms for the palevioletred module related to intracellular transport, mRNA metabolism, signaling, and chromatin remodeling. The bar plot shows the number of genes annotated to each term out of the 2416 genes in the module. GO term enrichment analysis was performed using g:Profiler with a q-value cutoff of 0.05. (**C**) Regulatory network showing predicted transcription factor (TF) regulators of the palevioletred module. The network was constructed using GENIE3, ARACNE, and CLR algorithms on gene expression data. The five transcription factors with highest connectivity (degree) are labeled. Node size corresponds to degree.

## Data Availability

The RNA-Seq datasets generated during this study are available in the NCBI Gene Expression Omnibus (GEO) repository, accession number GSE241954. All other data generated during this study are included in this published article and its Appendix A.

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
