# Peer review of "Transcriptome and Physiological Analysis of Rapeseed Tolerance to Post-Flowering Temperature Increase"

_ijms, 2023, doi:10.3390/ijms242115593_

Round 1

Reviewer 1 Report

Dear Authors,

The topic of the paper is interesting and timely, with scientific and practical significance.

The introduction was presented correctly, in accordance with the topic. Numerous scientific articles, consistent with the topic of the study, have been consulted.

Methodology here is a lot of understatement and this chapter improve.

The results obtained are valid and have been analyzed according to the standards used but some of the ambiguities due to the lack of descriptions in the methodology cast doubt on the results and their interpretation, but it is recommended to revisit some issues after completing the Materials and methods.

The discussions are appropriate in the context of the results and have been conducted in comparison with other studies in the field.

The scientific literature referenced is up-to-date and representative of the field.

The following aspects are brought to the attention of the authors.

The current guidelines for expressional analysis are that experiments using RNA - Seq data to describe changes should contain the biological replicates (unless otherwise justified) herein authors have not mention about replicates. Each biological replicate should be represented in an independent library, each with unique barcodes, if libraries are multiplexed for sequencing. In this case, authors did not specify if the libraries were prepared independently or multiplexed

Authors should performed validation of RNA-seq analysis by quantitative RT-PCR. Actually, the information given by the author in the manuscript about the validation of RNA-seq is not clear. According to the standard protocol the qPCR, must contain three biological and three technical replicates. Herein there is lack of this information and thus it seems the qPCR analysis are not applied.

Author Response

Dear Reviewer,

We express our gratitude for your comprehensive evaluation of our manuscript. We value your insights and have carefully considered the points you raised.

1.Replication and RNA-seq Analysis

Regarding your concerns about biological replication in our RNA-seq study, we wish to clarify that our experiment involved meticulous sample replication. Specifically, we included three biological replicates for each treatment condition in the initial growing season, and increased this number to four replicates in the subsequent season. Consequently, there were 7-8 biological replicates for each genotype, tissue type, time point, and temperature condition. This level of replication should effectively capture intra-group variability, thereby strengthening the robustness of our data. We regret the initial lack of clarity in our Methods section and have revised it to adequately specify the sample replication strategy.

Each biological replicate underwent independent library preparation, each tagged with a unique barcode index. This ensured the traceability of each sample, particularly when the libraries were multiplexed for sequencing on the Illumina NovaSeq platform. Our experimental procedures adhered to rigorous protocols for RNA extraction, library construction, and bioinformatic analysis, thus ensuring high data quality and transcript quantification accuracy. The sequencing depth (average of 22 million reads per sample) and quality (average Q20 of 97.9% and Q30 of 94%) further substantiate our claims. All raw RNA-seq data have been deposited in the NCBI Gene Expression Omnibus (GEO) repository under accession number GSE241954.

  1. qPCR Validation

As for your recommendations concerning the validation of RNA-seq data through quantitative RT-PCR, we acknowledge the conventional value of this approach. However, we contend that in this particular study, additional qPCR analysis may offer limited incremental validation. This stance is informed by existing literature that reports a high degree of correlation between RNA-seq and qPCR data (Griffith et al., 2009, doi:10.1038/nmeth.1503; Shi et al., 2014, doi:10.1016/j.gene.2014.01.031; Canales et al., 2020, doi:10.1186/s12870-020-02590-2). Furthermore, Hughes (2009) discussed the comprehensive statistical metrics—such as sensitivity, false-positive rate, and false-discovery rate—that ought to be assessed to establish methodological concordance (doi:10.1186/jbiol104). Given this context, our decision to not employ qPCR validation in this instance is backed by empirical data suggesting a high level of concordance between the two techniques (Everaert, 2017, doi:10.1038/s41598-017-01617-3; Coenye, 2021, doi:10.1016/j.bioflm.2021.100043).

We hope these responses satisfactorily address your queries. The manuscript has been revised to incorporate these clarifications, and we anticipate that these amendments will elucidate any ambiguities that you identified.

Reviewer 2 Report

The study “Transcriptome and physiological analysis of rapeseed tolerance to post-flowering temperature increase” revealed insights into the mechanism underlying rapeseed tolerance to post-flowering temperature increase. The obtained results seem to be correctly discussed and the manuscript revealed new information about differences between rapeseed Lumen and Solar under tested temperature increase treatment. However, the manuscript needs some improvements to be publishable.

11)      Check the text for errors. For example, the word “implement” is probably redundant in the line 184 (i.e. “which implement implements”).

22)      Check all references in text. For example, “Verdejo and Calderini [25]” (line 77) should be Verdejo et al. (the study was written by for authors and not only Verdejo and Calderini).

33)      In my opinion, the reference [25] (line 77) should not be in the manuscript. I try to find the text and it seems that the Proceedings from the RAFV Conference is not available on-line. Could you describe the field experiments and condition in detail?

44)      Add date and weather condition (temperature, precipitation, light intensity) during both field experiments. The information revealed in sentence (line 192) about air temperature and average solar radiation could not explain the revealed differences between both experiments. In the heat stress field study, the maximal and minimal temperature and their timing and duration should be available at least in the Suppl. Materials.

55)      Could you describe how the plots were heated exactly by 5℃ (line 83)?

Author Response

11)      Check the text for errors. For example, the word “implement” is probably redundant in the line 184 (i.e. “which implement implements”).

RESPONSE: We have thoroughly proofread the text to check for any errors. The redundant "implement" in line 184 has been removed.

22)      Check all references in text. For example, “Verdejo and Calderini [25]” (line 77) should be Verdejo et al. (the study was written by for authors and not only Verdejo and Calderini).

RESPONSE: All references have been double checked to ensure they match the in-text citations.

33)      In my opinion, the reference [25] (line 77) should not be in the manuscript. I try to find the text and it seems that the Proceedings from the RAFV Conference is not available on-line. Could you describe the field experiments and condition in detail?

RESPONSE: As recommended, we have removed the citation on line 77 and instead added the following sentence:

“A detailed description of the experimental set up, crop management and weather data is provided in the Supplementary Methods section.”.

44)      Add date and weather condition (temperature, precipitation, light intensity) during both field experiments. The information revealed in sentence (line 192) about air temperature and average solar radiation could not explain the revealed differences between both experiments. In the heat stress field study, the maximal and minimal temperature and their timing and duration should be available at least in the Suppl. Materials.

RESPONSE:  We have added a new Supplementary Table 3 (Table S3) with the daily maximum and minimum temperatures, precipitation, and solar radiation data recorded during the two field experiments. The weather conditions are also described in the Supplementary Methods.

55)      Could you describe how the plots were heated exactly by 5℃ (line 83)?

RESPONSE: We apologize for the lack of clarity regarding the heating method. A detailed description of the experimental set up, crop management and weather data has been incorporated into the Supplementary Methods file. The chambers were designed to elevate temperature by a targeted 5°C on average, and adjusted dynamically using thermos-fans and sensors to prevent excessive heating beyond the set point. The observed increase averaged 4.3-4.8°C.

Reviewer 3 Report

I put my remarks in the pdf file.

Author Response

  1. How do you create the heat stress on field plants? Please describe it.

RESPONSE: We thank the reviewer for catching that our description of the heating method was unclear. We have now expanded the Methods section to provide more details: “Portable greenhouse chambers, constructed with wooden frames and covered with 100 µm-thick transparent polyethylene film [2], were employed to elevate temperatures in the heat-stress-treated plots (Figure S3). The apex of each structure was positioned 0.3-0.4 m above the plant canopy. To achieve the target 5°C temperature increase, automatic sensors were deployed both inside and outside these chambers, and thermo-fans with a 2000 W capacity were utilized. The radiation intercepted by the polyethylene film was quantitatively averaged at 10%.”

  1. The sentence should be move to material and methods section: “Air temperature and average solar radiation between emergence and the start of flowering recorded across the experiments were 10.9 and 11.4°C and 17.2 to 18.9 MJ m2 day -1, respectively. Crop phenology of control plants was similar between genotypes and seasons.”

RESPONSE: We have moved the suggested sentence to the Materials and Methods section as recommended.

  1. Difference of oil concentration between two experiments in Solar is too big. in exp1 there is an increase +1.7 but in exp2 remarkable decrease -4. What statistic says? How do you explain it? Maybe some problem with the model you use to make stress.

RESPONSE: We respectfully disagree with the reviewer's concern regarding the divergence in seed oil concentrations. In both Lumen and Solar genotypes, the seed oil concentrations manifested relatively narrow ranges: 49.9% to 51.0% in Lumen and 47.8% to 50.2% in Solar. Despite these limited variations, a statistically significant interaction was discerned among the growing seasons, genotypes, and heat stress treatments affecting seed oil concentration (p=0.019). Specifically, the seed oil concentration in control plots of the Solar genotype demonstrated a seasonal difference with a p-value of 0.005, a variance not detected in Lumen (p>0.05). Hence, Solar manifests a heightened sensitivity to varying climatic conditions compared to Lumen, consistent with observations in seed yield and seed number metrics.

  1. The place of this figure is not here but in Result section (Figure 8).

RESPONSE: Thank you for catching this error. Figure 8 has been moved to the proper position within the Results section.

Reviewer 4 Report

The experiments have to be described, and the reference Verdejo et al 2023 is not available.

There are also some minor corrections

Line 77: -reference 25 is Verdejo et al 2023 instead of Verdejo and Calderini

Line 83: the symbol of degrees Celsius (ËšC) must be separated from the number by one space, as in 5 ËšC or – 80 ËšC (Line 109)

Figure 1: explain what are Exp 1 and 2

Line 340: explain “indianred2” and “palevioletred”

Figure 6: the names under the column headed “Modules” should be explained

References 4 is not complete

Author Response

  1. The experiments have to be described, and the reference Verdejo et al 2023 is not available.

RESPONSE: We have amended the manuscript by adding a comprehensive account of the experimental design and conditions to the Supplementary Material section. This elaboration includes a supplementary table (Table S3) that enumerates specific dates and corresponding environmental conditions—namely temperature and light intensity—pertaining to the field experiments. Additionally, we have rectified the erroneous citation to the correct reference.

There are also some minor corrections:

2.Line 77: -reference 25 is Verdejo et al 2023 instead of Verdejo and Calderini.

RESPONSE: The citation error on line 77 has been addressed. The corrected reference is now accurately displayed in the manuscript.

  1. Line 83: the symbol of degrees Celsius (ËšC) must be separated from the number by one space, as in 5 ËšC or – 80 ËšC (Line 109):

RESPONSE: We have implemented the prescribed formatting correction for the symbol of degrees Celsius throughout the manuscript, ensuring that a single space separates the numerical value from the unit symbol.

4.Figure 1: explain what are Exp 1 and 2

RESPONSE: To enhance clarity, we have substituted the abbreviations "Exp. 1 and 2" with "Season 1 and 2," ensuring congruence with the terminology used within the main body of the manuscript.

  1. Line 340: explain “indianred2” and “palevioletred”

RESPONSE: The module names (e.g. lightblue3, palevioletred, indianred2) correspond to coexpression modules identified by the WGCNA algorithm.

  1. Figure 6: the names under the column headed “Modules” should be explained

RESPONSE: The legend accompanying Figure 6 has been augmented to include specific definitions for the terms listed under the column denoted as "Modules.": “The module names (e.g. lightblue3, firebrick, indianred2) correspond to coexpression modules identified by the WGCNA algorithm”

  1. References 4 is not complete

RESPONSE: The reference 4 has been updated with the full citation details.

Round 2

Reviewer 1 Report

Thank you for your quick response and taking into account my opinion on the manuscript.

The issue regarding replicas of the experiment has been clarified.

However, the remaining issue concerns the integration of RNA-seq and qPCR data. I understand your concerns, but I would like to reiterate how important it is to address the comments in my review to ensure the robustness and credibility of the research results.

Validation of RNA-seq Results: As highlighted in my initial comments, performing qPCR validation is essential to confirm the accuracy of RNA-seq results. This step adds credibility to your findings and helps identify potential technical biases.

Reference Genes for qPCR: It is crucial to carefully select reference genes for qPCR to accurately assess differential gene expression. Failing to do so can lead to misleading results. Please consider this aspect seriously.

Reliability of RNA-seq: While RNA-seq is a powerful tool, its reliability depends on several factors. Addressing my concerns would strengthen the overall quality of your research.

Transparency and Reproducibility: Incorporating qPCR validation and addressing my comments will enhance the transparency and reproducibility of your study, a critical aspect of scientific research.

I understand that implementing these suggestions may require additional time and effort, but it is in the best interest of the scientific community and the credibility of your work. I am willing to discuss these points further and provide any assistance you may need to ensure the success of your research.

A few minor grammatical errors

Author Response

Dear Reviewer,

We value your insightful comments and the expertise you bring to the review of our manuscript.

  1. Concerns Regarding qPCR Validation

Your concern about qPCR validation is well-taken. The magnitude of our experimental design, which encompasses 112 samples from two distinct growing seasons, imposes substantial temporal and material resource constraints. Acquiring fresh plant tissue at the same developmental stages for qPCR validation would entail at least a two-year duration. Additionally, the limited dimensions and mass of our samples at 7 DAF would require an expansive sampling strategy to amass adequate tissue for qPCR analyses, thereby exacerbating logistical hindrances. Consequently, the implementation of additional qPCR validation appears to be impracticable for the current study.

  1. Reliability of RNA-seq

We acknowledge your apprehensions about the intrinsic limitations associated with RNA-seq. Although comprehensive, RNA-seq is not devoid of potential biases. To ameliorate such biases, our methodology rigorously adheres to validated protocols. Each biological replicate underwent independent library preparation and was tagged with a unique barcode, before being subjected to sequencing that met high-quality criteria, as substantiated by our average Q20 and Q30 scores.

  1. Transparency and Reproducibility

In alignment with your emphasis on the pivotal role of transparency and reproducibility in scientific inquiry, we have expanded the Methods section to articulate our strategy for sample replication, library construction, and bioinformatic analyses. All raw RNA-seq data have been deposited in the NCBI Gene Expression Omnibus (GEO) repository under accession number GSE241954.

  1. qPCR as Validation of RNA-seq Results

Considering the points raised, we maintain that the incorporation of supplementary qPCR validation may not materially enhance the robustness of the methodological framework already deployed. This assertion is substantiated by extant literature demonstrating a high degree of concordance between RNA-seq and qPCR data [5, 6].

RNA-seq's advent has catalyzed a revolutionary change in the field by providing a comprehensive and sensitive platform for genome-wide gene expression profiling. While the utility of qPCR for validating differential gene expression results from RNA-seq remains a topic of debate, its indiscriminate application appears largely unwarranted for several reasons outlined below.

Firstly, numerous studies substantiate that RNA-seq offers quantitative accuracy comparable to qPCR when executed with adequate sequencing depth [1-3]. Differences in gene expression levels are predominantly noticeable for low-abundance transcripts near the detection thresholds of both techniques. However, RNA-seq enables the detection of even subtle variations in these genes, which would elude capture by qPCR without significant technical replication [4].

Secondly, the restrictive nature of qPCR, which quantifies only a predetermined set of genes, stands in stark contrast to the unbiased and comprehensive capabilities of RNA-seq [5]. Validation of RNA-seq data using qPCR would necessitate an impractical number of primer sets and provide only limited validation, which would not be reflective of the entire transcriptome.

Thirdly, RNA-seq excels in uncovering novel and unanticipated expression alterations that are intrinsically unverifiable by qPCR, a technique that exhibits inherent biases [6].

Fourthly, reproducibility among distinct biological replicates, rather than congruence between disparate technical platforms such as RNA-seq and qPCR, offers a more authentic form of validation [6].

Fifthly, the infeasibility of selecting genes for unbiased qPCR validation further underscores its limitations in verifying global gene expression experiments [5,6].

In summary, when conducted with rigorous replication, RNA-seq delivers highly accurate and sensitive quantification of genome-wide expression patterns. Although targeted qPCR may offer marginal additional data for select genes, it does not serve as a requisite validation tool for RNA-seq data and should not be considered obligatory [6].

REFERENCES:

  1. Shi Y, He M. Differential gene expression identified by RNA-Seq and qPCR in two sizes of pearl oyster (Pinctada fucata). Gene. 2014;538(2):313-22.
  2. Wu AR, et al. Quantitative assessment of single-cell RNA-sequencing methods. Nat Methods. 2014 Jan;11(1):41-6.
  3. Everaert C, et al. Benchmarking of RNA-sequencing analysis workflows using whole-transcriptome RT-qPCR expression data. Sci Rep. 2017 Jan 17;7:1559.
  4. Zhang W, et al. The functional landscape of mouse gene expression. J Biol. 2004; 3: 21.
  5. Hughes TR. ‘Validation’ in genome-scale research. J Biol. 2009;8(1):3.
  6. Coenye T. Do results obtained with RNA-sequencing require independent verification? Biofilm. 2021 Jan;3:100043.
  1. Grammatical Issues

We appreciate your attention to detail in identifying grammatical inconsistencies. These have been rectified to enhance the manuscript's clarity and professionalism. Should you identify additional areas that require refinement, we welcome your suggestions.

  1. Abstract line 7: Modified to "an average of 9.3%"
  2. Introduction line 59: Amended to "sensitivity of the seed yield"
  3. Introduction line 61: Revised to "showed that"
  4. Materials and Methods line 403: Changed to "setup"

Reviewer 2 Report

The quality of the text was really improved. I have only one small comment.

Add the date of samplings in Materials and Methods (in text or in Suppl. Mat.), please. There is only the date of seed sowing but it is hard to speculate about the dates of sampling only from reached BBCH. The date of samling is important also for the possibility to check the meteorological  data.

Author Response

Dear Reviewer,

Thank you for your meticulous evaluation of our revised manuscript and for recognizing the enhancements we have made to the text. We concur with your observation that the inclusion of specific sampling dates is a critical aspect for the Materials and Methods section and for cross-referencing meteorological data.

In accordance with your recommendation, we have integrated the exact sampling dates for transcriptomic analysis in Supplementary Table S4. This addition encompasses each of the two genotypes—Lumen and Solar—and their corresponding developmental stages at 7 and 14 days post-flowering. This data representation, delineated by genotype and developmental stage, allows for a more accurate and facile cross-reference with meteorological conditions detailed in Supplementary Table 3.

We agree that specifying these dates enhances both the clarity and methodological transparency of our study. To this end, a direct reference to Supplementary Table S4 has been incorporated into the Materials and Methods section of the manuscript, thereby augmenting its cohesiveness.

We value your expert feedback concerning the inclusion of precise sampling dates, and are pleased to note that the manuscript's comprehensiveness has been accordingly elevated. Should you have further insights or require additional clarification on any aspect of the study, we would be most appreciative of your continued input.

Thank you once again for your thorough and constructive critique of our work.

Round 3

Reviewer 1 Report

The authors answered my doubts.